# First Evidence of “Earth Wax” Inside the Casting Molds from the Roman Era

**DOI:** 10.3390/molecules26144259

**Published:** 2021-07-13

**Authors:** Klára Jagošová, Jan Jílek, Pavel Fojtík, Ivan Čižmář, Miroslav Popelka, Ondřej Kurka, Lukáš Kučera

**Affiliations:** 1Department of Analytical Chemistry, Faculty of Science, Palacký University, 17. Listopadu 12, 779 00 Olomouc, Czech Republic; klara.jagosova01@upol.cz (K.J.); ondrej.kurka@upol.cz (O.K.); 2Section Classical Archaeology, Department of Archaeology and Museology, Faculty of Arts, Masaryk University, Joštova 220/13, 662 43 Brno, Czech Republic; jilek@phil.muni.cz; 3Institute of Archaeological Heritage Brno, Kaloudova 1321/30, 614 00 Brno, Czech Republic; fojtik@uapp.cz (P.F.); cizmar@uapp.cz (I.Č.); popelka@uapp.cz (M.P.)

**Keywords:** earth wax, ceresin, soot, X-ray fluorescence, gas chromatography, Roman era, mold, mass spectrometry, ion mobility

## Abstract

This research was focused on the analysis of material composition and organic residues present in three molds found in the Moravian region (Czech Republic) belonging to the Roman era. X-ray fluorescence spectroscopy pointed out the possible remelting of Roman objects in Barbarian territory. The analysis of organic residues retrieved from the internal part of mold #2 by pyrolysis-gas chromatography/mass spectrometry proved the presence of ozokerite wax (“earth wax”). Consequent analysis of this organic residue by Atmospheric Solids Analysis Probe–ion mobility spectrometry–high-resolution mass spectrometry (ASAP-IMS-HRMS) confirmed the presence of ceresin, the main component of ozokerite. Ceresin was also detected in a sample of the organic residue from mold #1. Note that this is the first application of ASAP-IMS-HRMS in archaeological research. The remains of earth wax in molds suggest the production of wax models as an intermediate stage for the production of lost-wax ceramic casting molds.

## 1. Introduction

The study of the production of non-ferrous metal objects in Central European Barbaricum is one of main tasks in the current research of the Roman era [1,2,3,4,5,6]. The recent chemical analysis of a metal mold belonging to Urnfield culture (1300–800/750 BC) proved the presence of carbon and beeswax in the inner part of the mold using infrared spectroscopy and gas chromatography/mass spectrometry (GC/MS). Authors hypothesize that carbon coating in the inner part of the casting mold could have been used as a layer preventing the mold from welding with the liquid metal poured into it [7]. Note that molds made of copper alloys are typical for the Barbarian region in the Roman era, where we register more findings than in the area of the former Roman Empire [8]. Another study, describing the usage of wax in metallurgical processes, was focused on GC/MS analysis of lost-wax ceramic casting molds in southern Levant [9]. This study revealed that not only animal waxes (such as beeswax), but also mineral ones were used. These waxes are commonly considered mineraloids (i.e., non-crystalline substances). The main representative of the group of mineraloids is ozokerite, also called “earth wax” [10,11]. This mineraloid can be found in bituminous deposits of Miocene age, close to oil-bearing deposits (including huge natural deposits in southeastern Poland and the northwestern Ukraine) [12]. Note that the extraction of ozokerite (specifically ceresin) is performed by boiling bituminous earth in water, and floating wax is collected [10]. The usage of wax in prehistory is well known, but its chemical detection in ancient artefacts is not so common, and the application of earth wax in prehistory was not detected at all. The main area in which wax materials were studied in more detail is art history. For that purpose, GC/MS [13,14,15,16], pyrolysis-GC/MS [13,17] and vibrational spectroscopy [12] were used.

The aim of this article is to perform detailed chemical analysis of three casting molds originating in the Roman era using pyrolysis-gas chromatography/mass spectrometry (Py-GC/MS), Atmospheric Solids Analysis Probe–ion mobility spectrometry-high-resolution mass spectrometry (ASAP-IMS-HRMS) and X-ray fluorescence spectrometry (XRF). To the best of our knowledge, there is no article dealing with the chemical analysis of organic compounds in casting molds from the Roman period. Moreover, this is the second evidence of application of Atmospheric Solids Analysis Probe-mass spectrometry in archaeological research, but it is being recorded for the first time in combination with ion mobility mass spectrometry.

## 2. Results and Discussion

Three molds from locations Klenovice na Hané (mold #1), Mikulov (mold #2) and Velké Hostěrádky-Dambořice (mold #3) were analyzed using XRF for the determination of their elemental composition (Figure 1). Table 1 shows that molds #1 and #2 contain high amount of Pb, i.e., 5.35% and 14.61%, respectively. Note that a similar chemical composition (high content of Pb) was also found in the case of objects related to metallurgy, i.e., a drainage channel of a damaged mold (10.25% of Pb) [18] and a mold of a knee-shape fibula (19.70% of Pb) [19]. In contrast, mold #3 contains only trace amounts of Pb (0.64%). This suggests that an alloy different from the Roman type was used in the melting process [18]. Lead was mainly used in the Roman Empire for the preparation of alloys for the production of statues, small statuettes, massive parts of toreutic products (e.g., attaches and protomes of bronze vessels) [1,6,20] and certain types of fibulae and their components [20,21]. The advantage of lead addition to alloys was the improved ”flowability“ of the metal, which provides homogeneous infilling of the mold [22,23]. However, in the Barbarian territory (Germania Magna), lead alloys were not very popular for the production of small metal objects (fibulae, belt fittings, etc.). High lead content in alloys was undesirable due to a higher risk of object damage during their consequent manual modification [6]. Considering the significant popularity of lead in the Roman Empire [6] and its addition to alloys (especially from the beginning of the second century to the fourth century AD [20]), we hypothesize that molds #1 and #2 were made by the remelting of objects of Roman origin [6,18]. The origin of alloy of mold #3 cannot be specified.

Subsequent experiments were focused on the analysis of the solid material from molds #1 and #2 (there was no infill present in mold #3) by pyrolysis-gas chromatography/mass spectrometry (Py-GC/MS) for the determination of organic compounds. Bonaduce and Colombini found long-chain alkanes indicating beeswax in their sample of the wax sculpture “The Plague” (1691–1694) by Gaetano Zumbo during their studies of works of art by Py-GC/MS [13]. A pyrogram of mold #2 also shows the presence of a higher amount of a long-chain alkane. Long-chain alkanes were found only in C14–C19 range—this could be caused by the pyrolysis of alkanes with longer carbon chains. Additionally, the detection of naphthalene in the chromatogram confirms the presence of organic compounds in the sample (Figure 2). Subsequently, the obtained pyrogram was evaluated using the F-Search program, version 3.6.3. (Frontier Lab, New Ulm, MN, USA). This program allows one to compare the pyrogram (i.e., its combined spectrum from all peaks) with the internal database of polymers in the program. The results point to the presence of ozokerite wax C1–C40 in mold #2 (Appendix A). Note that no long-chain alkanes were detected in pyrogram of mold #1.

The identification of detected compounds (long-chain alkanes) solely using an MS spectra database can be insufficient for the analysis of archaeological samples. For that reason, the ASAP-IMS-HRMS technique, allowing the measurement of the exact *m*/*z* value and drift time, was used. ASAP-IMS-HRMS data, obtained by a different ionization technique (at atmospheric pressure conditions), support the obtained results (Appendix A). The ASAP-IMS-HRMS technique (in the MS scan mode) detected a compound with *m*/*z* value 321.3122 that was identified as ceresine (deviation from the theoretical mass, dtm 3.5 mDa, C_22_H_41_O^+^). The presence of this compound strongly supports the identification of the source material as earth wax. The analysis of the ceresine standard provided the same peak profile in the mobilogram as the compound found in mold #2 (Figure 3A,B). Furthermore, fragmentation patterns of the ceresine standard and the compound correspond well with each other. The first fragment (*m*/*z* 303.2983, dtm 6.3 mDa, C_22_H_39_^+^) arises by the loss of water from the parent ion. The consequent fragments arise by cascade losses of C_2_H_4_ molecules (see Δ28 Da in spectra) (Figure 3C,D). Note that compound *m*/*z* 321.3181 (dtm—2.4 mDa) was also found in the sample from the mold #1 (Appendix A) and displayed the same fragmentation pattern as the ceresine standard. However, ceresine is present in a low concentration in this mold, which results in the lower intensity of its MS/MS spectrum (Figure 3D). We can, therefore, suppose that mold #1 contained ozokerite wax as well (Figure 3). To the best of our knowledge, this is the first application of the ASAP-IMS-HRMS technique for the analysis of archaeological samples.

The internal part of all three molds contained black pigment that was analyzed by Raman microscopy. The spectra reveal two major signals at 1328 and 1589 cm^−1^ (Figure 4). Sadezky et al. ascribe these very strong and broad signals to soot [24]. Raman analysis of standards of activated carbon confirms these results (Figure 4d).

Advanced chemical analysis of three molds from the Roman era provides important information on barbaric metallurgy. Baron et al. (2016) proposed that the presence of compounds related to wax in the mold points to the fact that the mold has never been used for casting [7]. However, we hypothesize that the remains of wax detected in molds #1 and #2 point to their use for the production of wax models. Such models could be used as an intermediate stage in the production of final objects, which were produced using lost-wax casting molds [25]. Note that carbon layers in the internal parts of molds allow the direct casting of metal objects [7].

## 3. Materials and Methods

### 3.1. Archaeological Samples

The research was focused on the analysis of three molds found in the Moravian region (Czech Republic) belonging to the Roman era. The first mold was found at the location Klenovice na Hané, Prostějov district (mold #1, length 57.0 mm, width 23.0mm, thickness 9.6 mm, stored in the Museum and Gallery Prostějov, inv. no. 310926, Uniform Trigononometric Cadastral Network, UTCN X: 1141119, Y: 552620). The mold was used for the production of strap ends of Raddatz O/12–14 type [26] and Madyda-Legutko type 2/6 [27]. This category of artifacts includes a wide range of strap ends, horse harness belts and sword belt fittings [28,29]. All products belonging to the Raddatz O group fall within the range of years 150/160–240/250 AD [30]. The second mold from the location Mikulov-Mušlov“, Břeclav district (mold #2, length 38.0 mm, width 45.0 mm, thickness 11.0 mm, stored in the Moravian museum in Brno, inv. no. 170560, UTCN X: 1205577, Y: 598456) represents a mold used to produce rings—a part of the Vimose-type bridle chain [28,31,32,33]. These harnesses were popular in the middle Danube region, especially in the second half of the 2nd century, and in the first decades of the 3rd century AD [33,34]. The third mold was found in the location Velké Hostěrádky-Dambořice at the area of Ždánický les, Břeclav and Hodonín district (mold #3; length 46.0 mm, width 30.0 mm, thickness 10.0 mm, stored in the Moravian Museum in Brno, inv. no. 170697, UTCN X: 179421, Y: 578652) This mold was used for the production of a barbaric knee-type fibula, type A 132. These products were popular in the territory of Germania Magna in the second half of the 2nd century AD [19,35,36,37].

### 3.2. X-ray Fluorescence Spectrometry (XRF)

A total of 100 mg of sample was taken from the intact core of each mold (three samples from each) using a 0.75 mm carbide drill and Proxxon Micromot 60/EF drill (Proxxon GmbH, Föhren, Germany). Elemental analysis of the samples was performed using an X-ray fluorescence spectrometer Vanta (Olympus, Southborough, MA, USA). The measurement parameters were as follows: analytical mode, excitation energy range: 8–40 kV, acquisition time: 310 s.

### 3.3. Pyrolysis-Gas Chromatography/Mass Spectrometry (Py-GC/MS)

Pyrolysis-gas chromatography/mass spectrometry was used for the determination of organic compounds in the solid infill of molds #1 and #2. Agilent 8890 GC system combined with Agilent 5977B MS system with Mass Hunter software (Agilent Technologies, Palo Alto, CA, USA) was used for the analysis. Separation was performed on UA5-30M-0.25F 20072108S Ultra Alloy (30 m × 0.25 mm × 0.25 µm) with a constant flow of 1.2 mL·min^-1^. Nitrogen (Messer Group GmbH, Bad Soden, Germany) was used as the collision gas with a flow rate of 1.5 mL·min^−1^. The initial oven temperature was 70 °C for 1 min; then, the oven was warmed up at the rate of 30 °C·min^−1^ to the value of 320 °C, which was held for 10 min; acquisition time: 19.3 min, inlet temperature: 300 °C, injection in split mode—ratio 50:1. The pyrolysis of 10 mg of the solid material was performed prior to analysis in an EGA/PY-3030D (Frontier Lab, New Ulm, MN, USA) pyrolysis unit at 550 °C for 0.5 min.

### 3.4. Atmospheric Solid Analysis Probe-Ion Mobility Spectrometry-High-Resolution Mass Spectrometry (ASAP-IMS-HRMS)

A Synapt G2-S (Waters, Torrance, CA, USA) mass spectrometer equipped with an Atmospheric Solids Analysis Probe, atmospheric pressure chemical ionization and ion mobility (ASAP-IMS-HRMS) was used for the untargeted analysis of individual samples taken from molds #1 and #2. A total of 5 mg of each sample was extracted by 300 μL of acetone (Penta, p.a., Czech Republic) in an ultrasound bath for 15 min. Consequently, the samples were centrifuged for 5 min at 5000× *g* RPM and liquid part was evaporated with a fine stream of nitrogen gas. Samples were dissolved in 100 μL of acetone. The glass stick was dipped into the individual acetone extracts. The same clean ASAP glass stick was used as a blank. The glass stick was subsequently fixed to the ASAP probe and inserted to the ion source. A new glass stick was used for each analysis, reducing the risk of the carryover effect. The probe was then gradually heated and desorbed compounds were ionized in a discharge. In order to increase the signal of analytes, approximately 2 μL of extract was loaded into the open-ended glass stick and analyzed in the same manner. The method parameters were as follows: ASAP Mode: positive, time of analysis: 3 min (initial probe temperature, PT: 400 °C, final PT: 600 °C), trap collision energy: 4 eV, transfer collision energy: 2eV, IMS wave velocity: 550 m·s^−1^, IMS wave height: 40 V, source temperature: 100 °C, sampling cone: 30 V, corona current: 2μA, corona voltage: 3 kV. Lock mass correction of the high-resolution mass spectrometer was performed using a leucine–enkephalin mixture (i.e., *m*/*z* 556.2771 in positive mode).

### 3.5. Raman Microscopy (RM)

The black particles found on the surface of molds were analyzed using a DXR2 Raman microscope (Thermo Scientific, Waltham, MA, USA). The parameters of measurement were the following: laser wavelength: 785 nm, laser power: 1 mW, aperture: 50 mm slit, collect exposure time: 2 s, amount of sample exposures: 16. A commercially available activated carbon (Sigma Aldrich, St. Louis, MO, USA) was used as the standard.

## 4. Conclusions

Three copper alloy molds from the Moravia region belonging to the Roman era were analyzed by XRF, Raman microscopy, Py-GC/MS and ASAP-IMS-HRMS. The elemental composition (i.e., higher content of Pb and Zn) of molds #1 and #2 points to the possibility of the production of “barbarian” alloys from remelted Roman objects. Mold #3 was made from pure copper, and therefore, it is not possible to further specify its origin. The internal part of all three molds contained a thin black layer that was identified as soot by Raman microscopy. Moreover, molds #1 and #2 contained solid material that was identified as a residue of ozokerite (earth wax). We hypothesize that the remains of earth wax in these molds point to their use for the production of wax models and the consequent use of these models for lost-wax ceramic casting. This first application of ASAP-IMS-HRMS to the analysis of archaeological samples brings new evidence on Barbarian metallurgy and the usage of earth wax in the past.

## Figures and Tables

**Figure 1 molecules-26-04259-f001:**
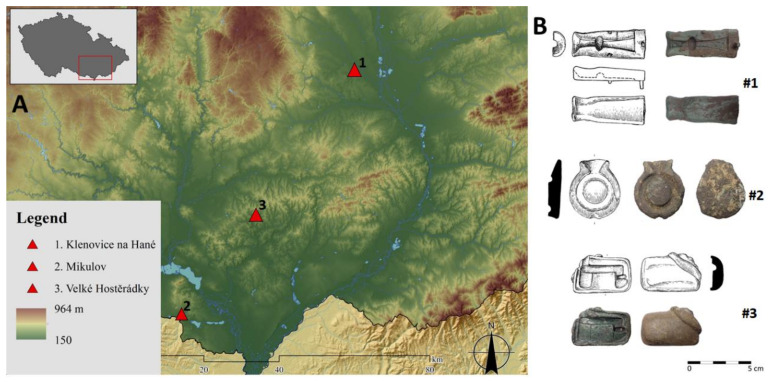
Map of the Moravian region with the locations of finds (**A**), photos and drawings of studied molds (**B**).

**Figure 2 molecules-26-04259-f002:**
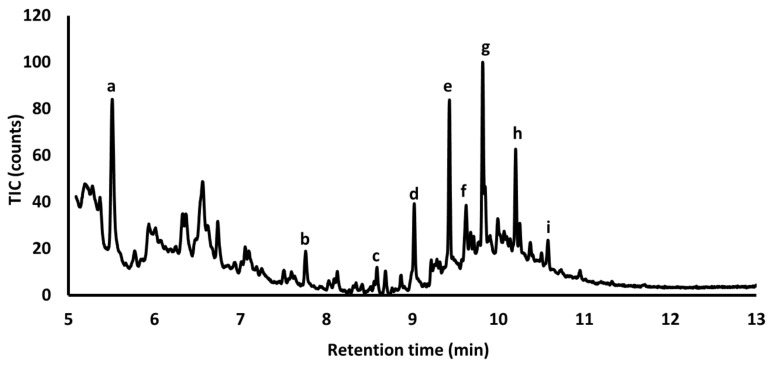
Pyrogram of the solid sample from mold #2 (a—furfural, b—naphtalene, c—tetradecane, d—pentadecane, e—hexadecane, f—2,6,10-trimethyl-pentadecane, g—heptadecane, h—octadecane, i—nonadecane).

**Figure 3 molecules-26-04259-f003:**
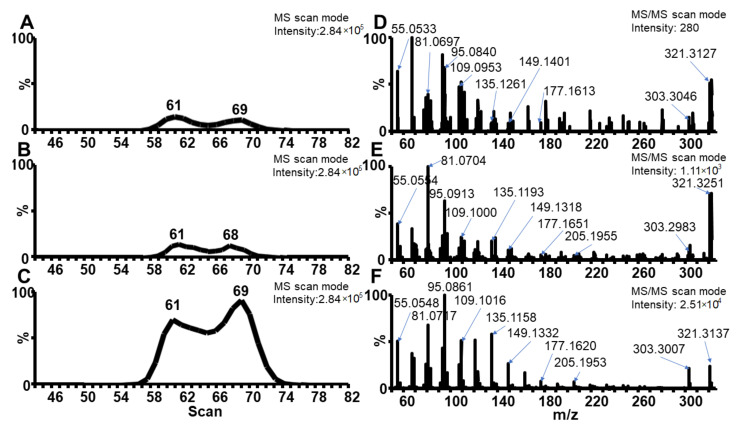
ASAP-IMS-HRMS extracted ion mobilograms (**A**–**C**) and ASAP-IM-MS/MS fragmentation spectra for *m*/*z* 321.316 (**D**–**F**), i.e., detected in mold #1 (**A**,**D**), mold #2 (**B**,**E**) and ceresin standard (**C**,**F**).

**Figure 4 molecules-26-04259-f004:**
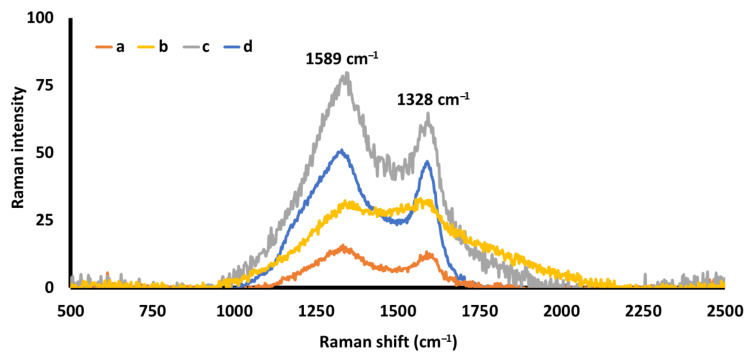
Raman spectra of the black solid material attached to the surface of molds #1 (a), #2 (b) and #3 (c), and activated carbon as the standard material (d).

**Table 1 molecules-26-04259-t001:** XRF analysis of alloy samples from the examined mold samples (avg—average; SD—standard deviation).

Element	Mold #1		Mold #2		Mold #3	
	avg	SD	avg	SD	avg	SD
Cu	88.64	3.62	79.33	4.15	98.17	0.70
Pb	5.35	2.33	14.61	2.93	0.64	0.25
Zn	2.82	0.06	1.62	0.06	0.00	0.00
Sn	2.69	1.55	3.97	1.21	0.56	0.36
Fe	0.37	0.19	0.13	0.02	0.20	0.09
Ni	0.06	0.01	0.05	0.01	0.05	0.01
Ti	0.04	0.03	0.08	0.00	0.04	0.03
Co	0.01	0.02	<LOD	-	<LOD	-
Au	<LOD	-	<LOD	-	<LOD	-
Zr	<LOD	-	0.01	0.00	<LOD	-
Cr	<LOD	-	0.01	0.00	0.01	0.01
Sr	<LOD	-	<LOD	-	<LOD	-
Nb	<LOD	-	<LOD	-	<LOD	-
Bi	<LOD	-	0.03	0.01	<LOD	-
Mn	<LOD	-	0.01	0.00	0.01	0.00
S	<LOD	-	<LOD	-	0.32	0.23
Ag	<LOD	-	0.15	0.11	0.00	0.00

## Data Availability

The data presented in this study are available on request from the corresponding author. The data are not publicly available due to the privacy policy of the author’s institution.

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
