# Peer review of "First Evidence of “Earth Wax” Inside the Casting Molds from the Roman Era"

_molecules, 2021, doi:10.3390/molecules26144259_

Round 1

Reviewer 1 Report

This approach to reconstructing Roman-era casting mold use is interesting, and the results are important and should be published. The paper requires much more background and explanation of the interpretations, however, and the conclusions should  be qualified. 

My own area of expertise is in GC/MS, and so my comments will focus on this portion of the paper. In areas outside my area of expertise, the XRF results seemed valuable and important, but the authors seemed rather too positive about their interpretation of recycled Roman metals...it is an interesting hypothesis, but requires some acknowledgement that it is a hypothesis, and not a firm conclusion.

In terms of the py-GC/MS, the results were interesting, but perhaps not has positive as the authors suggested. Most importantly, ozokerite would normally include truly long-chain alkanes, up to C32, and generally ranging from C20 to C32. The chromatogram shown in this paper ranges from C14 to C19, which is a bit short for ozokerite. If the authors are arguing that these alkanes are the product of pyrolysis or breakdown of the ozokerite, they should make the argument explicitly, and also explain why longer-chain alkanes were detected by Columbini and Bonaduce in their article on beeswax.

The authors seem to be basing their ozokerite identification very heavily on a database match with a known modern standard to argue that their compound is ozoerkerite, but these identifications are extremely risky when working with archaeological remains. Given the effects of assorted types of breakdown (hydrolysis, oxidation, and microbes) on even very stable compounds in the archaeological record, as well as potential contamination issues, modern standards tend to be sadly unreliable. The figure shown in S1 appears to be a broadly generic 'alkane' mass spectra...I'm afraid computer systems find matching these types of spectra extremely difficult because there is so much similarly in the base peaks, and the MW peaks are so small. As a result, I was not confident in the py-GC/MS and MS identification of ozokerite.

The presence of naphthalene is a potential line of evidence in support of the author's hypothesis--it should be explicitly mentioned. I was a bit surprised that there weren't more PAHs present, given the composition of ozokerite and the apparent presence of soot, but the presence of naphthalene is better than no PAHs.

More background was required on why 321 and 303 were such diagnostic peaks for the ASAP-IMS data--these peaks are clearly helpful in identifying ceresin, but it's also important to look into other compounds that might have these peaks. Are there any possible false positives? The authors did not appear to consider the possibility of other compounds being present with similar fragments. Unfortunately, such a possibility is almost always present when looking at archaeological remains.

In the end, this is an interesting paper that indicates the possibility of wax coating or casting in three molds from the Czech republic. The evidence for the origin of the wax is not as positive as suggested by the authors; more background and qualifications of the conclusions is needed before publication.

Author Response

The authors would like to thank to reviewer for his valuable work, which has greatly improved the manuscript. We have carefully considered the reviewers’ comments and have addressed them in a revised version of the manuscript. We have addressed specific comments and questions below.

Comment 1:

My own area of expertise is in GC/MS, and so my comments will focus on this portion of the paper. In areas outside my area of expertise, the XRF results seemed valuable and important, but the authors seemed rather too positive about their interpretation of recycled Roman metals...it is an interesting hypothesis, but requires some acknowledgement that it is a hypothesis, and not a firm conclusion.

Answer:

Authors agree with this comment. The conclusion dealing with suspected origin of alloys was written unambiguously. We modified this statement to the following version: “Elemental composition (i.e. higher content of Pb and Zn) of molds #1 and #2 points to the possibility of production of "barbarian" alloys from remelted Roman objects“

Comment 2:

In terms of the py-GC/MS, the results were interesting, but perhaps not has positive as the authors suggested. Most importantly, ozokerite would normally include truly long-chain alkanes, up to C32, and generally ranging from C20 to C32. The chromatogram shown in this paper ranges from C14 to C19, which is a bit short for ozokerite. If the authors are arguing that these alkanes are the product of pyrolysis or breakdown of the ozokerite, they should make the argument explicitly, and also explain why longer-chain alkanes were detected by Columbini and Bonaduce in their article on beeswax.

Answer:

Authors agree that results from py-GC/MS are ambiguous. However, the identification of long-chain alkanes in samples pointed to the need for more sensitive analysis by high-resolution mass spectrometry. Application of such technique combined with ion mobility separation is the highlight of article. Colombini and Bonaduce analysed beeswax in their samples (an animal wax), but in our case, we are focused to analysis of mineraloids (a mineral wax). For that reason, we only mentioned that long-chain alkanes pointed to presence of some waxy material. In main text we discuss the pyrolysis of longer alkanes to shorten ones (C14-C19).

Comment 3:

The authors seem to be basing their ozokerite identification very heavily on a database match with a known modern standard to argue that their compound is ozoerkerite, but these identifications are extremely risky when working with archaeological remains. Given the effects of assorted types of breakdown (hydrolysis, oxidation, and microbes) on even very stable compounds in the archaeological record, as well as potential contamination issues, modern standards tend to be sadly unreliable. The figure shown in S1 appears to be a broadly generic 'alkane' mass spectra...I'm afraid computer systems find matching these types of spectra extremely difficult because there is so much similarly in the base peaks, and the MW peaks are so small. As a result, I was not confident in the py-GC/MS and MS identification of ozokerite.

Answer:

Authors would like to thank reviewer for this comment. Authors agree that characterization of ozokerite (and wax generally) in archaeological sample performed only by comparison with NIST and Frontier databases is insufficient. For that reason, we used another technique allowing measurement of the exact m/z value (ASAP-IMS-HRMS). We mentioned it also in text: “Identification of detected compounds (long-chain alkanes) using chemical record database can be insufficient for analysis of archaeological samples. For that reason, new ASAP-IMS technique, allowing measurement of exact m/z value and drift time, was used.”

Comment 4:

The presence of naphthalene is a potential line of evidence in support of the author's hypothesis--it should be explicitly mentioned. I was a bit surprised that there weren't more PAHs present, given the composition of ozokerite and the apparent presence of soot, but the presence of naphthalene is better than no PAHs.

Answer:

Authors agree that detection of naphthalene in the chromatogram confirms presence of organic compounds in the sample. This sentence was added to the manuscript.

Comment 5:

More background was required on why 321 and 303 were such diagnostic peaks for the ASAP-IMS data--these peaks are clearly helpful in identifying ceresin, but it's also important to look into other compounds that might have these peaks. Are there any possible false positives? The authors did not appear to consider the possibility of other compounds being present with similar fragments. Unfortunately, such a possibility is almost always present when looking at archaeological remains.

Answer:

ASAP-IMS allows measurement of exact m/z values; this significantly reduces the possibility of the ions belonging to another compound (their deviation from the theoretical mass equals 3.5 and 6.3 mDa, for the 321 and 303 ions, respectively). The ion mobility separation allows to separate compounds based on their spatial arrangement. Obtained drift time of the analyte in our sample is identical with the authentic ceresin standard. Moreover, comparison of fragmentation spectra of both strongly supports its identification. Although there is always a chance that a different compound is present instead, in our opinion, the identification is based on strong results in this particular case.

Comment 6:

In the end, this is an interesting paper that indicates the possibility of wax coating or casting in three molds from the Czech republic. The evidence for the origin of the wax is not as positive as suggested by the authors; more background and qualifications of the conclusions is needed before publication.

Answer:

Authors disagree with this comment. The identification of ozokerite in molds was proved by two different instrumental techniques. ASAP-IMS-HRMS allows the measurement of exact m/z value. Obtained results show small deviation from theoretical mass, i.e. 3.5 and -2.4 mDa, and both the sample and the standard possess the same drift time in mobility separation.

Reviewer 2 Report

The article describes the archaeometric studies on material composition and organic residue in molds from Moravian region. However, the article is well constructed and is worth to publish, but a few of problems raises doubts and should be clarified. My recommendation is the minor revision of the text. I hope that proposed changes improve the value of the text.

General  comments:

The novelty of the described work should be better underlined, the scientific problem seems to be rather local, but the studies may be interesting for wider readership.

The validation of all analytical procedures (e.eg information about basic metrological parameters – detection limits, precision, estimation of the uncertainty and traceability studies (the details of results of CRMs analysis) should be added (e.eg. in Supplementary data).

The number of significant places (not decimal) for all data presented in the text should be according with the metrological rules based on the validation of the analytical methods.

Author Response

The authors would like to thank to reviewer for his valuable work, which has greatly improved the manuscript. We have carefully considered the reviewers’ comments and have addressed them in a revised version of the manuscript. We have addressed specific comments and questions below.

Comment 1:

The novelty of the described work should be better underlined, the scientific problem seems to be rather local, but the studies may be interesting for wider readership.

Answer:

The article is focused on analysis of three molds found in Central Europe, i.e. in the Barbarian territory. Due to the not availability of those rare artifacts from other countries we have to made a conclusion only for our region. In fact, the results could be different in other countries. However, the main novelty of the article is the first application of ASAP-IMS to analysis of an archaeological sample.

Comment 2:

The validation of all analytical procedures (e.eg information about basic metrological parameters – detection limits, precision, estimation of the uncertainty and traceability studies (the details of results of CRMs analysis) should be added (e.eg. in Supplementary data).

Answer:

Only qualitative analysis of samples was performed, so no validation of analytical method was done for consecutive quantification.

Comment 3:

The number of significant places (not decimal) for all data presented in the text should be according with the metrological rules based on the validation of the analytical methods.

Answer:

The lock mass correction of used high-resolution mass spectrometer was performed by leucine-enkephalin mixture, i.e. m/z 556.2771. This information was added to the manuscript.